# YOUR CLIP MODEL MIGHT BE UNDERTRAINED

## ABSTRACT

Contrastive Language-Image Pretraining (CLIP) models exhibit good performance on a range of vision tasks. To improve the performance of this class of models even further, several works have proposed to modify the CLIP training procedure. In this work, we show that it is possible to achieve substantial gains using a much simpler strategy. Specifically, existing CLIP models—especially those trained on smaller datasets—tend to be undertrained. Indeed, we show that extending the training procedure according to a simple heuristic can significantly improve the performance of CLIP models.

## 1 INTRODUCTION

In recent years, zero-shot inference has become a popular paradigm in the natural language processing (NLP) and computer vision communities. In this paradigm, one trains a model on vast amounts of data in order to learn generalizable features that then can be leveraged—without fine-tuning or weight updates—to perform inference on downstream tasks (Brown et al., 2020; Radford et al., 2021). Within the computer vision community, one popular approach to zero-shot image classification involves Contrastive Language-Image Pre-Training (CLIP) Radford et al. (2021), which gave rise to CLIP models with remarkable performance on a range of vision tasks, including ImageNet and several of its variants corresponding to various distribution shifts (Recht et al., 2019; Hendrycks et al., 2019; Barbu et al., 2019; Wang et al., 2019; Hendrycks et al., 2020). This successes let to a whole line of works that aim to further improve the performance of CLIP models (Yu et al., 2022; Mu et al., 2021; Li et al., 2022b; Wortsman et al., 2021).

**Our contributions.** In this paper, we propose a simple modification to the CLIP training procedure—specifically, restarting the learning rate schedule and training for a few extra epochs—and show that this significantly boosts the performance of CLIP models on several downstream tasks. In particular, we demonstrate that our approach alone leads to results that are competitive with previously proposed CLIP model performance improvements. This also suggests that existing CLIP models, especially the ones trained on smaller datasets such as Conceptual Captions 3M and 12M (Sharma et al., 2018; Changpinyo et al., 2021), might have been undertrained.

## 2 BACKGROUND

CLIP is a recent model with impressive performance on a variety of tasks (Radford et al., 2021). Previous work has argued that a key driver of CLIP's performance is its ability to leverage image-caption pairs during training (Santurkar et al., 2022). This ability, in turn, enables large-scale training on a vast amount of image-caption pairs scraped from the Internet, alleviating the need for (expensive and time-consuming) dataset annotation.

**Model architecture and zero-shot capability.** A CLIP model is composed of two networks: an image encoder $E_I$ (*vision backbone*, typically a ResNet (He et al., 2015) or a ViT (Dosovitskiy et al., 2021)) and a transformer-based text encoder $E_T$ (*text backbone*). Given an image-caption pair $(x, t)$, where $x$ is an image and $t$ is the corresponding textual caption, CLIP models encode the image $x$ and the caption $t$ using the relevant encoders into an image embedding $I$ and a caption embedding $T$, respectively, that belong to the same space. This design is well-suited for *zero-shot inference*, i.e., inference on new tasks without weight update. For example, by leveraging CLIP's ability to process textual prompts, one can map *unobserved* labels from new classification tasks into captions,

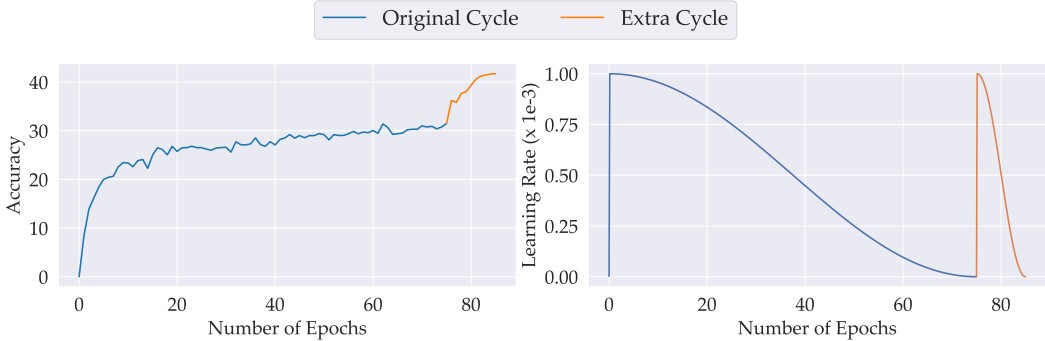

**Figure 1: CLIP models trained on smaller datasets are undertrained.** The blue curve on the left corresponds to the zero-shot accuracy of a CLIP model trained on CC12M for 75 epochs. This model was trained using the learning rate schedule represented by the blue curve on the right, and achieves a zero-shot ImageNet accuracy of 31%. After resetting the scheduler and training for 10 additional epochs (orange curve on the right), the zero-shot accuracy of the CLIP model on ImageNet increases by 10% (orange curve on the left).

then perform classification by matching each image embedding to its closest class embedding in the shared space.

**Performance.** CLIP models demonstrated impressive zero-shot performance on several image classification tasks, such as ImageNet classification (Deng et al., 2009), with recent large CLIP models (trained on LAION (Schuhmann et al., 2021)) surpassing 90% accuracy on ImageNet (Fang et al., 2023). Furthermore, unlike standard supervised classification models, CLIP models tends to be much more robust to distribution shift, including achieving high accuracy on the corresponding ImageNet variations benchmarks (Recht et al., 2019; Hendrycks et al., 2019; Barbu et al., 2019; Wang et al., 2019; Hendrycks et al., 2020).

## 3 Improving CLIP Training

Given the impressive performance of CLIP models, there was a number of works interested in further improving them. For example, several approaches proposed modifying the training objective (Mu et al., 2021; Li et al., 2022b), imposing additional supervision during training (Li et al., 2022a), leveraging additional data augmentations (Fini et al., 2023), or imposing a particular structure on the learned representations (Fürst et al., 2022; Goel et al., 2022). Sharing this goal, in this section we first investigate the performance of CLIP models trained on smaller datasets and show that these models might be undertrained. We then propose a simple modification to the training procedure that significantly boosts the performance of CLIP models. We finally investigate the effectiveness of the training procedure when applied at a larger scale.

### 3.1 CLIP Models Trained on Smaller Datasets are Undertrained

Given the significant cost of training CLIP models on large datasets such as LAION (Schuhmann et al., 2021), many researchers opt instead to train their CLIP models on smaller datasets such as Conceptual Captions 3M (CC3M) (Sharma et al., 2018) or Conceptual Captions 12M (CC12M) (Changpinyo et al., 2021). During training, the performance of these CLIP models saturates after few epochs, and simply training for longer does not significantly affect accuracy. For example, when training a CLIP model for 75 epochs on CC12M, the accuracy of the model has little improvement after the 40[th] epoch (see Figure 1).

However, we show that applying a very simple fine-tuning strategy after training can quickly boost accuracy. Specifically, when we "reset" the learning rate scheduler to its initial state (before training) and train for a small number of additional epochs, the performance of CLIP models improves significantly. For example, the zero-shot accuracy of a ResNet-50 CLIP model trained for 75 epochs on

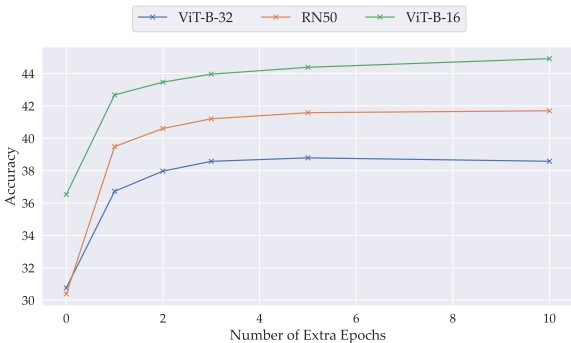

**Figure 3: Applying the additional training procedure for few extra epochs is enough to improve performance.** The ImageNet zero-shot accuracy of several CLIP models (y-axis) increases as we apply the additional training procedure for more epochs (x-axis). Note that the performance improvement saturates after applying the procedure for only three additional epochs. See Appendix B for additional results.

CC12M saturates around 31% on ImageNet. However, after applying our strategy for 10 additional epochs, the zero-shot accuracy of the resulting model jumps to 41% (see Figure 1).

We validate the effectiveness of our strategy by applying it to more CLIP architectures and measuring the performance improvement on several downstream tasks. We observe a consistent, significant improvement in several settings (see Table 2). This improvement suggests that the original CLIP models might have been undertrained.

| Model | ImageNet | ImageNet-V2 | ImageNet-A | ImageNet-O | ImageNet-R | ImageNet-Sketch | ObjectNet |
|-------|----------|-------------|------------|------------|------------|-----------------|-----------|
| ResNet-50 | 41.7 (+11.3%) | 35.6 (+9.84%) | 9.44 (+4.39%) | 47.5 (+8.15%) | 53.5 (+11.3%) | 31.1 (+8.55%) | 28.2 (+8.24%) |
| ViT-B-32 | 38.6 (+7.83%) | 33.1 (+7.14%) | 7.76 (+2.75%) | 43.9 (+7.70%) | 52.8 (+10.2%) | 31.6 (+7.95%) | 22.1 (+6.63%) |
| ViT-B-16 | 44.9 (+8.40%) | 38.0 (+7.09%) | 12.0 (+3.59%) | 45.5 (+5.60%) | 60.5 (+9.77%) | 35.1 (+8.17%) | 30.3 (+7.73%) |

**Table 2: Our simple training procedure consistently improves the performance of CLIP models trained on CC12M.** This table shows the zero-shot accuracy of several CLIP models on different downstream tasks after applying our simple strategy. The numbers in parentheses represent the absolute change in zero-shot accuracy on the corresponding downstream classification task. Note that the performance of several CLIP models improves significantly on each downstream task. For example, applying our simple strategy on a ResNet-50 CLIP model leads to a zero-shot accuracy of 41.7% on ImageNet—an improvement of 11.3% compared to the performance reported by the literature. See Appendix B for additional results.

### 3.2 How Many Extra Epochs Should We Train For?

As we have seen, resetting the learning rate scheduler and training for a fixed number of additional epochs improves CLIP performance. We now want to study how the zero-shot performance of several CLIP models changes as we vary the number of additional training epochs $K$.

To this end, we train three CLIP models (ResNet-50, ViT-B-32 and ViT-B-16), and then apply the extended training procedure to them, each time with a different number of additional training epochs. We observe that for models trained on CC12M, performance improvement saturates after applying three extra epochs (see Figure 3). This suggests that the extra overhead needed to achieve peak performance can be fairly minor.

### 3.3 At Which Epoch Should We Apply the Extended Training?

So far, we have been applying the extended training after the original model training was completed. However, would it be beneficial to apply it earlier in the training?

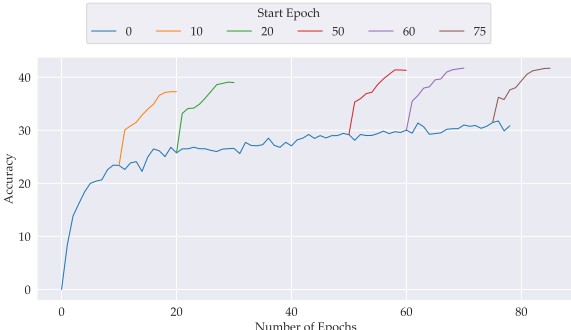

**Figure 4: Applying our strategy to improve performance.** The blue curve corresponds to the accuracy of the original CLIP model. Each other curve represents the zero-shot accuracy of the CLIP model after applying our strategy with different starting points. For example, the orange curve corresponds to applying our strategy on the CLIP model after it has been trained for 10 epochs (out of 75 epochs in total). Note that applying our strategy earlier during training leads to a performance improvement beyond the final accuracy reached by the model trained for 75 epochs. See Appendix B for additional results.

It turns out that such an earlier application can improve the model performance beyond what can be achieved with the full training cycle. To fully examine this phenomenon, we explore stopping the original training procedure at different epochs, and in each case we restart the training (and the learning rate scheduler) for 10 additional epochs. We observe that applying this extended training as early as 10 epochs into the original training cycle already improves performance beyond what is achieved after a complete training of the original model (see Figure 4). For example, our model reaches an accuracy of 37% after a total of only 20 epochs[1], higher than the original model's final accuracy of 31% that was trained on 75 epochs.

## 3.4 Using a Cyclic Learning Rate Can Improve CLIP Training

As mentioned in Section 3.1, CLIP models typically employ a single-cycle cosine learning rate scheduler (Loshchilov & Hutter, 2017) (see Figure 1). We showed, however, that 1) applying this schedule for a single cycle leads to suboptimal performance, and 2) employing an additional *short* cycle boosts accuracy. This additional cycle is reminiscent of cyclic learning rate schedulers. To investigate whether the use of such schedulers improves performance, we train a CLIP model using a multicycle cosine learning rate scheduler. Using this strategy, we obtain CLIP models that—with much fewer training epochs—outperform the CLIP models trained with the standard cosine learning rate schedule (see Figure 5).

## 3.5 Are Models Trained on Large-Scale Datasets Undertrained?

Our investigation thus far has revealed that CLIP models trained on smaller datasets might be undertrained. In this section, we investigate whether CLIP models trained on large-scale datasets, such as LAION-400M (Schuhmann et al., 2021), might be undetrained. To this end, we consider a CLIP model (with ViT backbone) pretrained on LAION-400M(Ilharco et al., 2021) and we apply the additional training procedure for 15 extra epochs (on the same dataset). Comparing the zero-shot performance of the new model to that of the original model, we observe that both models achieve similar performance on a range of datasets (see Table 6), suggesting that undertraining is less of an issue at scale.

## 3.6 Comparing with Existing Approaches

In the previous sections, we argued that CLIP models trained on smaller datasets might be undertrained. We also demonstrated that applying a simple additional training procedure can substantially

---

[1]The first 10 epochs correspond to the original training cycle, while the second 10 epochs correspond to our strategy.

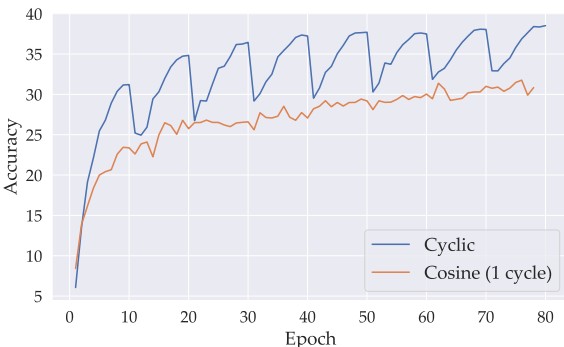

**Figure 5: Applying a cyclic learning rate schedule improves performance.** Each curve represents the ImageNet zero-shot accuracy of a ResNet-50 CLIP model as a function of the number of training epochs. The orange curve corresponds to the standard training strategy using a cosine LR scheduler, while the blue curve corresponds to training the CLIP model with a cyclic LR Scheduler. Note that applying a cyclic LR improves performance. See Appendix B for additional results.

|  | ImageNet | ImageNet-A | ImageNet-O | ImageNet-R | ImageNet-Sketch | ImageNet-V2 | ObjectNet |
|---|---|---|---|---|---|---|---|
| OpenCLIP | 62.94 | **21.69** | 53.45 | 73.40 | 49.39 | **55.11** | **43.91** |
| Ours | **63.29** | 19.80 | **56.05** | **75.51** | **52.82** | 54.84 | 43.09 |

**Table 6: CLIP models trained on large datasets are less likely to be undertrained**. Comparison of the performance of two ViT-B-32 CLIP models on several downstream tasks. The first row corresponds to a public CLIP model trained on LAION 400M (Ilharco et al., 2021), while the second row corresponds to applying our strategy to the public model. Note that training for an extra cycle leads to similar performance, which suggests that CLIP models trained on large datasets are less likely to be undertrained.

improve the performance of these models. In this section, we compare this strategy with other approaches employed to improve the performance of CLIP models.

Specifically, we consider several previously proposed methods for refining the basic CLIP training procedure (Chen et al., 2022; Goel et al., 2022; Fürst et al., 2022; Li et al., 2022a; Fini et al., 2023). These approaches either modify the training objective, or impose additional structure on the learned representations. The CLIP models trained from scratch on the smaller datasets using the proposed approaches show an accuracy improvement compared to baseline CLIP models. Our results demonstrate that simply applying the additional training procedure can provide competitive results (see Table 7).

| Method | Pretraining | |
|---|---|---|
|  | **CC3M** | **CC12M** |
| CLIP (baseline) (Radford et al., 2021) | 20.6 | 36.5 |
| ProtoCLIP (Chen et al., 2022) | 21.5 | – |
| CyCLIP (Goel et al., 2022) | 22.1 | – |
| CLOOB (Fürst et al., 2022) | 24.0 | – |
| DeCLIP (Li et al., 2022a) | 27.2 | 41.0 |
| CLIP (Improved) (Fini et al., 2023) | 27.4 | 44.4 |
| CLIP (Ours) | 24.2 | 41.7 |

**Table 7: Comparison of different CLIP training strategy.** The first row displays ImageNet zero-shot accuracy of ResNet-50 CLIP model trained using the canonical training scheme. Subsequent rows display the performance of CLIP models trained according to different strategies in the literature. Last row presents the performance obtained by applying our additional training strategy. Note that the various approaches in the literature improve the performance of the original model.

## 4 RELATED WORK

Zero-shot inference is an increasingly popular paradigm where the goal is to solve a specific task that is unseen during training Radford et al. (2021). One notable example of zero-shot models is the CLIP model (Radford et al., 2021) that achieved state-of-the-art results on a range of zero-shot classification tasks, most notably ImageNet and its variants (Recht et al., 2019; Hendrycks et al., 2019; Barbu et al., 2019; Wang et al., 2019; Hendrycks et al., 2020).

Given the remarkable performance of CLIP models, several works have proposed approaches to improve their performance. Some of these works modify the underlying training procedure. For example, FLIP masks at random a subset of the ViT input tokens and drops them (Li et al., 2022b). Other works have proposed using strong and weak data augmentations (for both vision and text backbones) when computing the similarity between an image and a caption (Fini et al., 2023).

Another line of work modifies the objective function used for training the CLIP model. For instance, SLIP (Mu et al., 2021) combines an image contrastive loss (SimCLR (Chen et al., 2020)) with the CLIP objective function (Radford et al., 2021). Other approaches propose clustering similar images and captions ((Chen et al., 2022)), imposing geometrical consistency between an image-caption pair embeddings ((Goel et al., 2022)), imposing additional supervision ((Li et al., 2022a)) or using Hopfield networks to control the covariance of the learned representations (Fürst et al., 2022).

Finally, additional works proposed pretraining the vision backbone prior to training the CLIP model (Fang et al., 2022; 2023; Sun et al., 2023), or ensembling several trained CLIP models (Ilharco et al., 2021).

## 5 CONCLUSION

In this paper, we have demonstrated that CLIP models trained on smaller datasets might be under-trained. To improve the performance of such CLIP models, we propose a simple additional training procedure, and demonstrate its effectiveness and competitiveness with existing approaches. This suggests that the methods proposed to improve CLIP performance should be tested at a larger scale in order to accurately reflect their potential benefits.

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
