# OpenReview forum: "Your CLIP Model Might Be Undertrained"
_ICLR.cc/2024/Conference — Submitted to ICLR 2024_

### Official Review · Reviewer_N6Pn · 2023-11-01

**Soundness:** 4 excellent
**Presentation:** 3 good
**Contribution:** 3 good
**Rating:** 6
**Confidence:** 5

**Summary:**

This paper makes a simple but intriguing observation that vision-language models trained on even a small dataset such as CC12m can improve substantially (10% zero-shot accuracy), by restarting the training of the model on the *same* data. The paper starts by showing this for on CC12m by training for an extra 15 epochs (Figure 1), then it shows that similar improvements are reached with 2 other models for various number of extra epochs (Figure 3), then shows that the improvements are consistently significant on various OOD datasets (Table 2), then they confirm that restarting training even from early checkpoints can result in faster training (Figure 4), then they extend the idea by repeating the restart by revisiting the idea of cyclical LR schedules and show that a cyclical LR can reach a higher accuracy than a single cycle cosine LR schedule (Figure 5). Finally, they use a cyclical LR schedule to train on the larger LAION400M dataset and conclude that “CLIP models trained on large datasets are less likely to be undertrained.”

**Strengths:**

- The results are surprising and promising for future directions on alternative LR schedules for multi-modal training.
- The paper is easy to read and asks natural questions sequentially that engages the reader. Although the reader is left with many unanswered questions by the end!

**Weaknesses:**

- While the paper makes interesting observations, the paper is missing a lot of discussion and potential for extending the observations given the unused page limit. A few unanswered, immediate and simple questions: How many cycles are optimal? How long should each cycle be? Is there a relation between the number of samples and the length of the LR cycles? Figure 5 only presents results with one schedule.
- Results in Figure 4 and 5 are limited to only one architecture and are not shown to hold for other architectures. Would these results hold for ViT-L/14?
- The final conclusion is that “CLIP models trained on large datasets are less likely to be undertrained.”. This is based on only one LR schedule and one model that does not provide definitive evidence for the conclusion.

**Questions:**

- Where are Figure 2 and Table 1?

---

> ### Author Response · Authors · 2023-11-22
>
> We thank the reviewer for their constructive feedback. We respond below to the raised questions.
>
> 1. *While the paper makes interesting observations, the paper is missing a lot of discussion and potential for extending the observations given the unused page limit...*
>
> We thank the reviewer for their suggestions. We have indeed experimented with some of them.
>
> *Re- number of extra cycles:* We have found in our investigation that additional cycles do not provide any further performance boost.
>
> *Re- length of cycle:* We investigated in Figure 3 how changing the cycle duration affects performance. We can see that a small cycle of 3 epochs (on CC12M) was enough to get most of the gain.
>
> *Re- relation with dataset size:* We have observed that for small datasets, 10 epochs was enough for both CC3M and CC12M (a dataset with 4x more samples). In particular, by applying a cycle with 10 epochs, we improve the ImageNet zero-shor accuracy of a ResNet-50 CLIP model by 4% when trained on CC3M (Table 7), and by 11% when trained on CC12M (Table 2).
>
> 2. *Results in Figure 4 and 5 are limited to only one architecture and are not shown to hold for other architectures. Would these results hold for ViT-L/14?*
>
> Our experiments on smaller datasets with several models suggest that the phenomenon is model-agnostic. For example, as we can see in Table 2, the ImageNet zero-shot accuracy of several CLIP models improves when applying our cycle: ResNet-50 by 11.3%, ViT-B-32 by 7.83% and ViT-B-16 by 8.4%. This improvement translates to other downstream tasks. We expect the same results with a ViT-L/14.
>
> 3. *The final conclusion is that “CLIP models trained on large datasets are less likely to be undertrained.”. This is based on only one LR schedule and one model that does not provide definitive evidence for the conclusion.*
>
> We agree with the reviewer that more analysis needs to be done at large scale to confirm. Unfortunately, such an analysis is beyond our compute capabilities, especially given the rebuttal period length. Our results from a single run on LAION-400M indicate only a little improvement by adding the extra training cycle, and as such, we think that applying our method to other large models would lead to the same results. We are happy to add more large-scale experiments for the final version if the reviewer thinks it would strengthen the paper.
>
> 4. *Where are Figure 2 and Table 1?*
>
> We thank the reviewer for pointing this out. We have a unified numbering system for figures and tables. We will fix that.

---

> ### Comment · Reviewer_N6Pn · 2023-11-22
> **I thank the authors for their response. My concerns remain unresolved.**
>
> I thank the authors for their response. My concerns remain unresolved.
>
> **W1** The paper is missing a lot of discussion and potential for extending the observations.
> I do not see a revision to the paper although the general response suggests. As such, my response is based on only the authors’ latest comments and other reviews.
>
> **W1.1** How many cycles are optimal?
> > We have found in our investigation that additional cycles do not provide any further performance boost.
>
> This does not answer my question about “optimal number of cycles”. Figure 4 and 5 suggest that the optimal number is more than 1 and possibly more than 8 (maximum in the plot) for ResNet50. My question is, a complete study on the optimal number of cycles even for ResNet50 is missing. More broadly, a study for various architectures is missing.
>
> **W1.2** How long should each cycle be?
> I agree Figure 4 partially answers this question. However, the following sentence is not concrete and the exact amount of drop needs to be reported and a plot that shows cost-vs-accuracy and connecting the final accuracy of each curve in Figure 4 would be important to include.
> > …a small cycle of 3 epochs (on CC12M) was enough to get most of the gain.
>
> **W1.2** Is there a relation between the number of samples and the length of the LR cycles?
> > We have observed that for small datasets, 10 epochs was enough for both CC3M and CC12M
>
> Although these two datapoints are important, they do not answer my question. I particularly asked for a study on the relation between the number of samples and the length of the LR cycle. Such a study can be done by varying the number of samples for a single dataset and finding the optimal LR cycle length. The suggested numbers do not provide a complete picture. Adding to this picture, the reduced gap on the large LAION dataset shows that there will be a saturation point in the plot I’m suggesting.
>
> **W2** Results in Figure 4 and 5 are limited to only one architecture and are not shown to hold for other architectures.
> > Our experiments on smaller datasets with several models suggest that the phenomenon is model-agnostic. For example, as we can see in Table 2 …
>
> I agree that Table 2 provides a partial evidence for the potential of the alternative LR schedule. However, here I’m particularly asking about Figure 4 and 5 which investigate shorter training and cyclical learning rates. This weakness also relates to W1 as the previously suggested analysis might provide different optimal values for different architectures.
>
> **W3** …does not provide definitive evidence for the conclusion.
> > such an analysis is beyond our compute capabilities, especially given the rebuttal period length
>
> I understand the compute limitations in general. However, limited results would weaken the applicability of any method and the conclusions.

---

### Official Review · Reviewer_B4x4 · 2023-11-01

**Soundness:** 2 fair
**Presentation:** 3 good
**Contribution:** 2 fair
**Rating:** 5
**Confidence:** 5

**Summary:**

This paper presents a training technique using which the performance of CLIP like vision language models can be improved without undergoing any change in objective function or model architecture. Authors of this work shows that CLIP like models trained on small scale datasets like CC-3M or CC-12M might be under trained. To improve their performance, simply finetuning a pre-trained CLIP model again with additional few epochs with a restarted learning rate scheduler is enough. The paper additionally shows that this technique is less effective when tried on CLIP models trained on large scale datasets like LAION-400M. The paper shows additional ablations and result comparisons with previous methods to provide a broad perspective.

**Strengths:**

1. This work shows that hyper-parameter and tweaking training strategies for large-scale pretraining of vision-language models plays significant roles in determining their performance on downstream tasks after training.
2. Restarting the LR scheduler with few-additional epochs is a simple and elegant way to improve performance of CLIP models trained on small scale datasets.
3. The resulting performance of the model is competitive to prior methods that bring additional objective function or model architecture changes to baseline CLIP model.
3. Most importantly, this study underlines the crucial need of bench-marking at larger scales to truly reflect improvements due on proposed modifications. Otherwise, showing effects on small scale datasets could be sometimes misleading.
4. The paper conducts fair comparison and additional ablation studies to provide a broad perspective about CLIP training.
5. Paper is easy to read and well presented.

**Weaknesses:**

1. In my view, this paper presents a effective but somewhat hyperparameter training technique which is more of a engineering trick rather than pure novel contribution. In other words, this paper says that one should use a altered version of multi-cycle LR scheduler instead of a single-cyle LR schedule to improve CLIP performance.
2. There is little or no analysis on why the proposed trick helps improve CLIP performance. This work can be further supplemented by conducting a detailed analysis on the learned embeddings, for example analysis on modality gap [1], t-SNE visualizations etc.
3. This work shows performance comparison on zero-shot tasks, but it will be great to see how the learned embeddings provide benefits for adaptation tasks like linear probing or its use on downstream tasks which uses CLIP features in their framework.
4. Although competitive to prior methods trained on CC3M and CC12M, it is unclear how this technique performs when combined on pre-trained models of prior methods. For example, does this technique shows complementary effect when plugged on CyCLIP pretrained model? (using same CyCLIP objective functions even in the proposed technique)
5. Similar results with-and without this technique on large scale CLIP models like CLIP-LAION400M shows that this trick is only valid for small scale models with less than 50% baseline accuracy.
6. There are missing tables in the paper. For example, I cannot see Table 1 anywhere in the paper.

**Questions:**

My main concern is that this paper mainly shows a training trick, which poses questions for novelty of this work. Please refer to the weaknesses section for my additional concerns and queries.

In summary the paper is nice but it lacks solid technical contributions which can be an significant issue.

---

> ### Author Response · Authors · 2023-11-22
>
> We thank the reviewer for their constructive feedback. We respond below to the raised questions.
>
> 1. *In my view, this paper presents a effective but somewhat hyperparameter training technique...*
>
> We fully agree with the reviewer that the method is a simple and straightforward baseline for CLIP. In fact, we have re-named the paper and positioned the method accordingly. Given its competitive performance, we believe it provides a better (and fast to compute) baseline for CLIP on smaller datasets.
>
> 2. *This work shows performance comparison on zero-shot tasks, but it will be great to see how the learned embeddings provide benefits for adaptation tasks like linear probing or its use on downstream tasks which uses CLIP features in their framework.*
>
> We thank the reviewer for these suggestions, and will incorporate the suggested experiments in our final paper. Sadly, we were not able to finish running all these experiments by the time of the rebuttal deadline, but we have some preliminary results that we believe are quite promising. Please refer to **[GP4]** for more zero-shot results and **[GP5]** for linear probing results.
>
> 3. *Although competitive to prior methods trained on CC3M and CC12M, it is unclear how this technique performs when combined on pre-trained models of prior methods...*
>
> We thank the authors for their suggestion. We have indeed experimented a bit with this idea and found out that applying the extra training cycle on top of other approaches leads to the same results as applying the extra training cycle by itself. Although we were not able to run conclusive enough experiments to update the draft of our paper, we are happy to include these results in the final version of the paper.
>
> 4. *Similar results with-and without this technique on large scale CLIP models like LAION400M shows that this trick is only valid for small scale models with less than 50% baseline accuracy.*
>
> We agree with the reviewer that the finding does not translate to large datasets. However, we believe that the smaller data regime is definitely interesting. Training CLIP models on large datasets is expensive and only large companies and a very limited number of academic labs afford it. Most academic labs however cannot train CLIP models except on smaller datasets. As such, when such works propose methods to improve the performance of CLIP models, it is necessary to consider better baselines to validate the effectiveness of the proposed method, and to increase our confidence that it would work on large datasets.
>
> We believe our approach can provide such a baseline given its simplicity.
>
> 5. *There are missing tables in the paper. For example, I cannot see Table 1 anywhere in the paper.*
>
> We thank the reviewer for pointing this out. We were using a unified numbering system for figures and tables (so Figure 1, Table 2, Figure 3, etc.), but will change it for the final version.

---

### Official Review · Reviewer_1NTN · 2023-11-01

**Soundness:** 2 fair
**Presentation:** 3 good
**Contribution:** 2 fair
**Rating:** 3
**Confidence:** 4

**Summary:**

The paper revisits the training schedule of the CLIP, especially on those trained on smaller-scale datasets, and finds that a simple continuing training with LR rewinding can significantly improve the CLIP baselines. It demonstrates improvements on 6 ImageNet variants with R50/ViT-B-32/ViT-B-16 backbones and even outperforms some approaches that are designed to be data-efficient (e.g. DeCLIP) when trained on CC12M.

**Strengths:**

- The finding of the paper is interesting
- It establishes a stronger baseline for CLIP training, and questions whether we should solely evaluate data efficient approaches on smaller scale, and its transferability to large-scale datasets.

**Weaknesses:**

- The finding does not necessarily transfer to models that are trained on large-scale datasets (e.g. 400M).
- There is not much analysis / theory on why such behavior exists (or not) on different scales of datasets.
- It would be interesting to see how the proposed schedule helps when we apply to other approaches in Table 7.

**Questions:**

See weaknesses.

---

> ### Author Response · Authors · 2023-11-22
>
> We thank the reviewer for their constructive feedback. We respond below to the raised questions.
>
> 1. *The finding does not necessarily transfer to models that are trained on large-scale datasets (e.g. 400M).*
>
> We agree with the reviewer that the finding does not translate to large datasets. However, we believe that the smaller data regime is still interesting (see general point **[GP1]**). Training CLIP models on large datasets is expensive and only large companies and a very limited number of academic labs afford it—most academic labs cannot train CLIP models except on smaller datasets. As such, when such works propose methods to improve the performance of CLIP models, it is necessary to consider better baselines to validate the effectiveness of the proposed method, and to increase our confidence that it would work on large datasets.
>
> We believe our approach can provide such a baseline given its simplicity.
>
> 2. *There is not much analysis / theory on why such behavior exists (or not) on different scales of datasets.*
>
> We refer the reviewer to **[GP2]** above—although more theory CLIP models are known to require a lot of training data to achieve a good performance. We believe that when the training data is limited, it is easy for the model to be stuck in a local optimum. As the dataset size increases, the sheer number of training data points leads to a better generalization.
>
> 3. *It would be interesting to see how the proposed schedule helps when we apply to other approaches in Table 7.*
>
> We thank the authors for their suggestion. We have indeed experimented a bit with this idea and found out that applying the extra training cycle on top of other approaches leads to the same results as applying the extra training cycle by itself. Although we were not able to run conclusive enough experiments to update the draft of our paper, we are happy to include these results in the final version of the paper.

---

> > ### Comment · Reviewer_1NTN · 2023-11-23
> >
> > I thank the authors for the additional clarification. My concerns are not fully addressed.
> >
> > *Given that the author's response was provided only a day before the end of the discussion period, it may be hard for the authors to provide additional feedback, but I will still provide my comments below.*
> >
> > > R3. We have indeed experimented a bit with this idea and found out that applying the extra training cycle on top of other approaches leads to the same results as applying the extra training cycle by itself.
> >
> > Can the authors please clarify what is "the same results" and what is "by itself"? It would be clearer and more direct if the authors can directly provide a table of the results: base method / original performance / performance after the additional schedule.
> >
> > And if the author's implication is that the other approaches do not benefit from the additional schedule, it would be more interesting to give an analysis of what is the reason for that -- what is fundamentally different between the original CLIP and the improved approaches, and why the additional schedule does not help.

---

### Official Review · Reviewer_MMUs · 2023-11-02

**Soundness:** 3 good
**Presentation:** 1 poor
**Contribution:** 2 fair
**Rating:** 3
**Confidence:** 4

**Summary:**

This paper reports the observation that existing popular CLIP training recipe is suboptimal on smaller datasets (under training), and demonstrates clear improvement by resuming the training with a higher learning rate. Experimental results on ImageNet show the effectiveness of the proposed modified training recipe.

**Strengths:**

The finding of the paper is clear and simple. The experiments seem convincing. It is a meaningful contribution and the community should move away from the current suboptimal CLIP recipe especially on the CC3M data.

**Weaknesses:**

1. Why is the proposed observation of under-training only present with the smaller dataset e.g. CC3M but not with the larger ones like LAION-400M? It'd be very helpful if the authors can provide some insights here, because this seems to be the core contribution of the paper. For example, does the noise level in the image-text dataset affect the fitting behavior in some way?

2. Many existing CLIP recipes rely on high-resolution finetuning after the low-resolution pretraining (e.g.ALIGN, CoCa), which is essentially doing the same finetuning with the extra cycle as shown in Figure 1 (right). Would the proposed approach still benefit models that are already trained with high-res finetuning?

3. For ablation, I think it'd be cleaner to compare with a baseline that is trained on the same number of epochs so that the only difference is the learning schedule (instead of adding additional training epochs).

4. The paper writing can use some improvement to expand on the introduction, method, analysis, and related work (still plenty of space left).

**Questions:**

See weaknesses in order.

---

> ### Author Response · Authors · 2023-11-22
>
> We thank the reviewer for their constructive feedback. We respond below to the raised questions.
>
> 1. *Why is the proposed observation of under-training only present with the smaller dataset e.g. CC3M but not with the larger ones like LAION-400M?...*
>
> We refer the reviewer to the general points **[GP2]** above. In short, additional experimentation rules out factors such as label noise or model capacity as the underlying causes of undertraining. These results suggest that the root of undertraining might simply be the number of data points (e.g., the model is more likely to be stuck in some local optimum).
>
> 2. *Many existing CLIP recipes rely on high-resolution finetuning after the low-resolution pretraining (e.g.ALIGN, CoCa)...*
>
> Good observation! As far as we know, fine-tuning on higher resolution images is usually employed with large datasets (where under-training is not an issue). At small scale, we suspect that it would provide similar benefits, provided that the learning rate scheduler and optimizer state are reset.
>
> 3. *For ablation, I think it'd be cleaner to compare with a baseline that is trained on the same number of epochs...*
>
> We thank the reviewer for their recommendation. We have trained several baseline models, on different number of epochs, ranging from 30 to 200, on CC3M and CC12M. We have observed that all these models have almost the same downstream performance at the end of their training.
>
> 4. *The paper writing can use some improvement to expand on the introduction, method, analysis, and related work (still plenty of space left).*
>
> We thank the author for their suggestion. We have indeed significantly revised the paper to include more details.

---

### Author Response · Authors · 2023-11-22
**General Response**

We thank the reviewers for their constructive feedback. We make a few clarifying general points in this comment, and then respond to each reviewer individually below.

**[GP1] Message**

We want to make it clear that the goal of our paper is not to introduce a new state-of-the-art method for training CLIP models. Instead, our key observation is that many modifications to the CLIP training procedure rely on small datasets to establish an improvement over the baseline CLIP procedure. Our contribution is (a) to show that this baseline is excessively weak due to undertraining for small datasets (b) to propose a procedure for mitigating the effects of this undertraining.

**[GP2] Why are only small datasets undertrained?**

 Even though our main message does not hinge on this, this is a good question that many reviewers asked—indeed, our work finds that undertraining is largely a problem for smaller datasets like CC3M and CC12M, but much less of a problem for large datasets like LAION-400M. Although a theoretical analysis of this phenomenon is beyond the scope of our work, we have added some experiments that try to uncover the root of this behavior. Specifically, we found that undertraining *cannot* be explained by either label noise or by model capacity, through the following experiments:

*Label Noise:* Before training our CLIP model on CC3M, we pre-tokenize the captions, and drop at random 10% of the tokens. We then train our CLIP model on this new data using a standard training procedure, and we observe that the final model ImageNet zero-shot accuracy remains 20%. We repeat the same experiment with dropping 20% of the tokens rather than 10%, and we observe that the model accuracy drops to 10%.

*Image Noise:* Similar to the previous setup, we add to all the images in CC3M before training random gaussian noise, and then we train our models. We observe that the final model accuracy remains close to 20%.

*Model Capacity:* We train our model with 2 different weight decays: 0.1 and 0.5. Both setups lead to models with a similar accuracy on ImageNet (20%).

After ruling these out, our current hypothesis is that in the small-data regime, CLIP models are more prone to reaching poor local minima, and the restarting process is enough to escape these local minima by increasing the stochasticity of SGD. We are happy to further test this hypothesis in the final version of our paper, if it would be of interest to reviewers.

**[GP3] Additional changes**

In our revision, we also have the following additional experiments
Improved general exposition and added discussion
Changed the title of the paper to “A Simple and Effective Baseline for CLIP”
Changed Figure and Table numbering from unified to independent numbering.
We also investigated the effect of using more than a single cycle. Our experiment indicates that a single cycle was sufficient to mitigate undertraining, and additional cycles did not present any benefit.
We add zero shot evaluation of our CLIP models on more downstream tasks (check **GP4**).
We provide results from linear probing on downstream tasks, and we observe similar gains (check **GP5**).

---

> ### Author Response · Authors · 2023-11-22
> **General Response (Continued)**
>
> **[GP4] Zero-Shot performance on more downstream tasks**
>
> We apply our procedure to CLIP models trained on CC12M, and we measure the zero-shot accuracy on several downstream tasks.
>
> | Validation Dataset | RN50                                   | ViT-B-32                                | ViT-B-16                                |
> |--------------------|---------------------------------------|-----------------------------------------|-----------------------------------------|
> | ImageNet           | 41.7 (*+11.3%*)                        | 38.6 (*+7.83%*)                        | 44.9 (*+8.4%*)                        |
> | ImageNet-A         | 9.44 (*+4.39%*)                       | 7.76 (*+2.75%*)                         | 12.0 (*+3.59%*)                         |
> | ImageNet-R         | 53.5 (*+11.3%*)                     | 52.8 (*+10.2%*)                        | 60.5 (*+9.77%*)                        |
> | ImageNet-S         | 31.1 (*+8.55%*)                       | 31.6 (*+7.95%*)                        | 35.1 (*+8.17%*)                        |
> | ImageNet-v2        | 35.6 (*+9.84%*)                      | 33.1 (*+7.14%*)                        | 38.0 (*+7.09%*)                        |
> | ObjectNet          | 28.2 (*+8.24%*)                      | 22.1 (*+6.63%*)                         | 30.3 (*+7.73%*)                        |
> | Aircraft           | 2.67 (*+1.05%*)                        | 2.31 (*+0.15%*)                         | 2.52 (*+0%*)                         |
> | CIFAR10            | 49.4 (*+18.5%*)                      | 81.3 (*+9.69%*)                       | 80 (*+2.06%*)                       |
> | CIFAR100           | 27.5 (*+13.0%*)                       | 43 (*+2.15%*)                         | 48.2 (*+4.09%*)                          |
> | Caltech101         | 76.4 (*+6.03%*)                     | 77.3 (*+3.39%*)                        | 79.1 (*+3.69%*)                        |
> | Flowers            | 34.6 (*+11.1%*)                     | 34.0 (*+10.6%*)                        | 37.8 (*+14.1%*)                         |
> | Pets               | 62.0 (*+13%*)                      | 57.8 (*+2.8%*)                        | 64.7 (*+12.2%*)                        |
> | SUN397             | 47.5 (*+2.93%*)                      | 47.3 (*+2.47%*)                         | 48.6 (*-0.6%*)                        |
> | Stanford Cars      | 26.2 (*+13.3%*)                      | 19.2 (*+7.26%*)                        | 26.8 (*+9.03%*)                        |
>
>
>
> **[GP5] Linear probing results on several downstream tasks**
>
> We apply our procedure to CLIP models trained on CC12M, and we measure the linear probing accuracy on several downstream tasks.
>
> | Validation Dataset | RN50 | ViT-B-32 | ViT-B-16 |
> |--------------------|------|----------|----------|
> | ImageNet           | 58.52 (*+4.77%*) | 56.88 (*+6.41%*) | 63.78 (*+4.85%*) |
> | ImageNet-A         | 4.56 (*+1.07%*)  | 4.13 (*+0.7%*)   | 6.73 (*+0.94%*)  |
> | ImageNet-R         | 45.61 (*+10.54%*)| 45.89 (*+8.79%*) | 54.77 (*+10.85%*)|
> | ImageNet-S         | 29.02 (*+9.02%*) | 29.99 (*+8.89%*) | 36.56 (*+10.52%*)|
> | ImageNet-v2        | 47.25 (*+4.39%*) | 45.54 (*+4.88%*) | 52.37 (*+5.01%*) |
> | Aircraft           | 48.48 (*+7.98%*) | 41.97 (*+2.16%*) | 48.54 (*+7.92%*) |
> | Birdsnap           | 41.76 (*+9.25%*) | 32.49 (*+4.53%*) | 42.9 (*+6.01%*)  |
> | CIFAR10            | 77.1 (*+8.22%*)  | 81.09 (*+3.48%*) | 81.55 (*+4.18%*) |
> | CIFAR100           | 54.34 (*+9.78%*) | 57.36 (*+5.05%*) | 55.7 (*+5.73%*)  |
> | Caltech101         | 94.42 (*+3.7%*)  | 93.32 (*+1.84%*) | 95.63 (*+1.77%*) |
> | Caltech256         | 83.45 (*+5.67%*) | 82.49 (*+4.46%*) | 86.49 (*+2.88%*) |
> | Flowers            | 92.57 (*+3.78%*) | 89.77 (*+3.79%*) | 93.36 (*+2.26%*) |
> | Food               | 77.09 (*+7.26%*) | 71.84 (*+4.98%*) | 80.44 (*+4.88%*) |
> | Pets               | 82.47 (*+7.05%*) | 80.76 (*+3.98%*) | 86.29 (*+5.78%*) |
> | SUN397             | 70.35 (*+4.62%*) | 67.44 (*+4.05%*) | 72.6 (*+3.71%*)  |
> | Stanford Cars      | 70.17 (*+12.09%*)| 60.61 (*+11.88%*)| 71.63 (*+9.88%*) |

---

### Meta-Review · Area_Chair_bTkq · 2023-12-12

**Metareview:**

This paper explores the training schedule of CLIP models, especially on smaller-scale datasets. The authors identify that continuing training with learning rate rewinding can significantly enhance performance. This simple yet effective method is verified on various ImageNet variants across different visual backbones. While the reviewers find this paper interesting to read, they raise several major concerns: 1) it lacks in-depth analysis or theory explaining why this training method is effective; 2) its effectiveness cannot transfer well to larger-scale datasets; and 3) the overall presentation of this paper can be improved. The rebuttal is considered, but reviewers are not fully convinced. The final decision is rejection.

The AC encourages the authors to carefully tackle these concerns and make a stronger submission next time.

**Justification For Why Not Higher Score:**

The reviewers have significant concerns about this paper and do not believe it is ready for publication at this time.

**Justification For Why Not Lower Score:**

N/A

---

### Decision · Program_Chairs · 2024-01-16

Reject